# Blood Metabolite Profiling of Antarctic Expedition Members: An ^1^H NMR Spectroscopy-Based Study

**DOI:** 10.3390/ijms24098459

**Published:** 2023-05-08

**Authors:** Laura Del Coco, Marco Greco, Alessandra Inguscio, Anas Munir, Antonio Danieli, Luca Cossa, Debora Musarò, Maria Rosaria Coscia, Francesco Paolo Fanizzi, Michele Maffia

**Affiliations:** 1Department of Biological and Environmental Science and Technology, University of Salento, Via Lecce—Monteroni, 73100 Lecce, Italy; laura.delcoco@unisalento.it (L.D.C.); marco.greco@unisalento.it (M.G.); ale.inguscio@unisalento.it (A.I.); anas.munir@studenti.unisalento.it (A.M.); antonio.danieli@unisalento.it (A.D.); lucagiulio.cossa@unisalento.it (L.C.); debora.musaro@unisalento.it (D.M.); 2Department of Mathematics and Physics “E. De Giorgi”, University of Salento, Via Lecce—Arnesano, 73100 Lecce, Italy; 3Institute of Biochemistry and Cell Biology, National Research Council of Italy, Via P. Castellino 111, 80131 Naples, Italy; mariarosaria.coscia@ibbc.cnr.it

**Keywords:** hypoxia, winter-over, Antarctica, Concordia base, adaptation, NMR, metabolomics

## Abstract

Serum samples from eight participants during the XV winter-over at Concordia base (Antarctic expedition) collected at defined time points, including predeparture, constituted the key substrates for a specific metabolomics study. To ascertain acute changes and chronic adaptation to hypoxia, the metabolic profiles of the serum samples were analyzed using NMR spectroscopy, with principal components analysis (PCA) followed by partial least squares and orthogonal partial least squares discriminant analyses (PLS-DA and OPLS-DA) used as supervised classification methods. Multivariate data analyses clearly highlighted an adaptation period characterized by an increase in the levels of circulating glutamine and lipids, mobilized to supply the body energy needs. At the same time, a reduction in the circulating levels of glutamate and *N*-acetyl glycoproteins, stress condition indicators, and proinflammatory markers were also found in the NMR data investigation. Subsequent pathway analysis showed possible perturbations in metabolic processes, potentially related to the physiological adaptation, predominantly found by comparing the baseline (at sea level, before mission onset), the base arrival, and the mission ending collected values.

## 1. Introduction

The human species, which originated on the African continent, gradually colonized all other continents except Antarctica [1]. However, the colonization of mountainous environments presented a great challenge, and it is estimated that the highest elevations of our planet, the Tibetan and Andean highlands were only colonized about 25,000 and 11,000 years ago, respectively, through profound physiological adaptations [2].

Indeed, as the altitude increases, there is a progressive reduction in atmospheric pressure inducing a condition of hypobaric hypoxia. The subsequent depletion of circulating blood oxygen (O_2_), because of its low concentration in lungs, triggers a state of hypoxemia in the body. As O_2_ is a key player in the energy generation process, acting as the final electron acceptor during oxidative phosphorylation process (OXPHOS) and several enzymatic reactions, the onset of hypoxemia triggers several adaptive responses in the body, aimed at restoring homeostasis. In addition to altitude, numerous other conditions can generate these responses, which include gas inhalation, use of anesthetics and pathologic conditions such as emphysema, pneumonia, pneumothorax, and asthma. Such circumstances can give rise to acute forms of hypoxia that become chronic when the body is unable to reverse the triggering condition. In both early and late hypoxia stages, however, the organism has a number of compensatory mechanisms, which are put in place [3,4].

An early adaptive response is enacted within seconds of a *P*_O2_ reduction. This effect is essentially due to the carotid bodies sensors, which modulate cardiac activity and ventilation, triggering a system-wide smooth muscle vascular response [5]. Subsequently, the persistence of hypoxia causes an increase in circulating erythrocyte number and in their ability to bind O_2_ due to higher hemoglobin levels, accompanied by a neovasculogenic process. These systemic responses are supplemented by cellular ones, as all O_2_-consuming cells possess mechanisms capable of rapidly detecting alterations in their supply and reacting accordingly [6].

A mitochondrial activity perturbation is one of the first consequences of pO2 alteration, which compromises OXPHOS while promoting glycolytic metabolism (Pasteur effect inhibition). This is followed by a reduction in ATP levels, an increase in reactive oxygen species (ROS) and an opening of mitochondrial ion channels, triggering a Ca^++^ ion release. This set of events acts as the first hypoxic sensor for the cell (mitochondrial response), which reacts by modulating its protein activity [7,8]. If the condition is sustained, a modulation in gene expression occurs with a proteome switch and a metabolic adaptation. These transcriptional changes, coupled with a metabolic flux modulation, indicate a specific metabolic rearrangement associated with hypoxia. Altogether, these variations mainly fall under the control of the master hypoxic response regulators, hypoxia-inducible factors (HIFs) [9,10].

The understanding of the cellular mechanisms triggered by O_2_ deficiency is a topic of great interest and relevance. Worldwide, millions of people develop pathological conditions associated with a moderate or severe hypoxic state, with numerous attributable deaths and disability-adjusted life years (DALYs) lost, particularly because of numerous respiratory diseases. Yet, at the same time, populations that live at altitudes ≥1500 m above sea level (ASL) do not exhibit any clinical problems attributed to chronic hypobaric hypoxia and, instead, often exhibit a reduced incidence of chronic diseases such as diabetes or cardiovascular disorders [11].

Studying hypoxia in individuals moving to high altitudes for prolonged periods of time is an ideal way to collect information about not just early responses and metabolic adaptations but also to distinguish hypoxic physiological reactions from those related to pathological states. Such a study requires the removal of certain variables that could represent a bias, for instance, underlying cardiovascular and pulmonary conditions, altitude variations and prolonged exposure to harsh temperatures, malnutrition, and dehydration [12,13]. Moreover, the possibility of acquiring biological data prior to the departure and during the stay, at well-defined time-points, is also of great relevance.

The Antarctic Italian–French base “Concordia” provides such an ideal environment where all these requirements are met. Located on the Antarctic Dome C plateau, 3233 m ASL, the base houses a small contingent of researchers for a full year in a mildly hypoxic environment. Despite the geographic altitude, the atmospheric pressure on the plateau is lower than that measured at the same height elsewhere on the planet (478 mmHg). In addition, the O_2_ fraction of is between 20.82 and 20.9%, while in other parts of the globe, it is 21% [14]. This is because the ocean currents that reach Antarctica are deep rather than shallow, thus less rich in O_2_ [15]. The two elements combined place the base at a virtual altitude of 4800 m ASL. Furthermore, an average temperature of −58 ± 9 °C, 41 ± 10% humidity, and a condition of permanent darkness during the austral winter, lasting for several months, greatly restrict outside activities [16].

The objective of this study was to examine changes in the metabolic cycles involved in acute and late response to prolonged mild hypobaric hypoxia following acclimatization. Metabolomics is an experimental approach capable of intercepting variations in circulating biomarkers, resulting from the alteration of cells metabolism and small molecular substrates, thus providing a chemical fingerprint attributable to a specific condition reflected by the state of its metabolic cycles. Nuclear magnetic resonance spectroscopy (NMR) is used in such a context and provides several advantages, such as an easy, nondestructive specimen preparation and a high reproducibility of the results [17,18].

To ascertain acute changes and chronic adaptation to hypoxia, we gathered serum samples from eight participants during the XV winter-over at Concordia base at seven defined time points, including predeparture (T0), base arrival (TI), and mission ending (TVI). The serum samples were then analyzed using ^1^H-NMR, which was combined with a principal components analysis (PCA), supervised partial least squares, and orthogonal partial least squares discriminant analyses (PLS-DA and OPLS-DA). This paper reports the results of a metabolomics study based on serum samples and with the purpose of understanding the acute changes and chronic adaptation of the human organism to hypoxia.

## 2. Results

### 2.1. NMR Analysis of Serum Samples

The current study focused on the potential differences of metabolic profiles of eight participants at the Concordia base during the winter-over, under nonpathological moderate hypoxemic hypoxic conditions, which enabled the collection of information on metabolic changes resulting from the adaptation process. A typical ^1^H-NMR spectrum of a serum sample is shown in Figure 1 (with the main metabolite NMR assignments described in Appendix A). As widely reported in the literature for the human blood [19], the NMR spectra composition is dominated by several patterns of signals, referred to as metabolites, such as amino acids, glucose, glycerol, lactate, citrate, and catabolic by-products (creatinine), together with a substantial number of hydrophobic or lipid-like molecules (triglycerides, lipoproteins). In particular, signals of lipids and lipoprotein components were assigned, with consideration to the terminal (-CH_3_ methyl groups at 0.90 ppm), long-chain (-(CH_2_)*_n_* methylenic at 1.30 ppm), allylic (CH_2_-CH=CH- at 2.01 ppm), and olefinic (CH=CH at 5.32 ppm) groups of fatty acids containing moieties of the various lipoprotein particles (especially HDL, LDL, and VLDL) [19]. Moreover, *N*-acetyl glycoproteins (GlycA/GlycB) resonances were assigned as broad signals at 2.05/2.07 ppm and further identified with the 2D ^1^H-^13^C HSQC spectrum, as shown in Appendix A. These signals refer to -COCH_3_ acetyl groups of glycoproteins (N-acetylglucosamine and *N*-acetylgalactosamine and *N*-acetylneuraminic acid, Appendix A). In particular, as shown by other authors, the presence of these signals are mainly due to the contributions of α1-acid glycoprotein (major contributor) followed by those of α1-antitrypsin [20].

### 2.2. Multivariate Analysis of NMR Data

After the preprocessing of the NMR spectra (phasing, baseline correction, bucketing procedure; see Materials and Methods Section 4.4), multivariate data analysis was applied to the binned data. Although the NMR spectra appeared similar among different serum samples, there were striking differences in peak intensities for the different groups, identified as initial sampling session (T0) and subsequent samplings after 7 (TII), 14 (TIII), 90 (TIV), 180 (TV), and 270 (TVI) days after Antarctic base arrival. Preliminarily, an unsupervised PCA was applied to the whole data, giving rise to some observations in terms of general data grouping (Appendix A). In particular, a time-dependent clustering was observed on the first principal component (PC1; with 27.8% of the total variance), with T0 samples at negative values of PC1 and TI samples in the middle of the score plot, while the other samples lay at positive values of PC1, with a scattering on the second principal component (PC2; with 14.7% of the total variance). Subsequent PLS-DA, performed according to the sampling session class, confirmed the group separation (Appendix A). The shift of metabolite profiles for the considered classes was clearly observed according to the sampling progression, especially along the first component (with 26.3% of the total variance). From these analyses, relatively higher levels of glutamate and *N*-acetyl glycoproteins (mainly GlycA) as well as a relatively lower content of lipids and glutamine, were found in samples collected at the initial sampling session (T0) as compared to those at the other time points. Moreover, univariate analyses (one-way ANOVA with the HSD post hoc test and repeated measures one-way ANOVA) were performed to confirm the statistical significance of the distinctive variables (selected peak related bins). The differentiated levels of the reported metabolites (normalized concentrations) are clearly represented on their associated box plots (Figure 2A–D). Results also showed that the perturbations, potentially related to the physiological adaptation, lead to statistically significant differences on response time (Appendix A). It is, therefore, possible to observe a time-dependent reduction in circulating levels of glutamate and GlycA following arrival at Concordia base. A relatively higher level of circulating lipids was also observed in comparison to the T0, which increased from the second measurement (TI) and tended to remain steadily prominent from the TIII onward. On the other hand, the levels of circulating glutamine showed an initial increase during the earliest hypoxic phase, with a small decrease in mean values at the TIII and a further reduction from the TIV to the end of the stay.

The PLS-DA model on the whole data showed sharp changes for the considered sampling classes from the basal time point T0 to TI and TVI, while a certain degree of smooth change in metabolites was observed from TI to TVI. Furthermore, considering all the pairwise combinations of the T0, TI, and TVI time points (Figure 3A–C), the resulting two-class OPLS-DA models showed a good class separation (R^2^X > 0.25, R^2^Y > 0.99, Q^2^ > 0.79 in all cases) (Appendix A). For each OPLS-DA model, the corresponding VIP (variable importance in projection) plot reports the relative contribution of the variables (the binned NMR signals) to the whole model. The principal discriminant metabolites responsible for class separation (VIP values > 2.0) were selected and used for further data analyses. We therefore observed a relative high content of glutamate, glucose, choline, and DMA (dimethylamine) at T0 with respect to TI and a relatively lower content of glutamine and lipoproteins, such as LDL and VLDL. The comparison between samples at T0 and TVI showed a relatively higher level of some metabolites, such as glutamate and GlycA, in the former time point. Indeed, glutamate and GlycA levels were noticeably reduced at the end of the winter-over, contrarily to what has been observed for 3-hydroxybutyrate and lipoproteins. Finally, comparing the TI with the TVI data, glucose and phenylalanine levels exhibited a relative increase, while 3-hydroxybutyrate and tyrosine showed relatively lower levels in the latter measurement.

The pairwise analyses on serum samples comparing T0 and TI (Figure 4A and Appendix A), T0 and TVI (Figure 4B and Appendix A), and TI and TVI (Figure 4C and Appendix A) allowed for the detection of some metabolic pathways potentially altered in a significant manner. Glutamine/glutamate and alanine/aspartate/glutamate metabolisms as well as arginine biosynthetic pathway appeared perturbed according to the comparison of both TI and TVI with respect to T0. A possible perturbation in phenylalanine/tyrosine/tryptophan biosynthesis and phenylalanine metabolism was also observed when comparing TI and TVI.

### 2.3. Measure of Glucose Levels

The absolute value of glucose content in each serum sample was evaluated colorimetrically. According to the spectrophotometric reading, the mean ± SEM glucose value of the subjects for each time point was calculated (Appendix A). Moreover, the ERETIC 2 (electronic reference to access in vivo concentrations) Bruker methodology was also used as direct quantification of blood glucose with the NMR spectra. A good correlation was obtained in terms of the concentration (mg/dL) calculated from the NMR spectra (see Section 4) and the colorimetric assay (Appendix A). The results showed high inter-individual nonsignificant variability, mainly due to the small number of study participants although this was within the normal range for healthy subjects. The amount of glucose in the sample remained stable during the first week at the base, with values not dissimilar to those obtained at T0 (100 ± 5.8 mg/dL). The glucose concentration exhibited a subsequent decline in the samples obtained two weeks after exposure to the hypoxic condition (90 ± 4.6 mg/dL). Values at TIV reverted to levels comparable with TI; they thus showed a slight, nonpathologic increase at TV (105 ± 4.5 mg/dL), the highest value of the entire winter-over. Finally, concentrations returned to basal levels at TVI.

## 3. Discussion

The study of the effects of hypoxia on the human body and, consequently, the comprehension of the biological responses elicited to counteract or adapt to it represent a challenge and a major goal for modern medicine. A hypoxemic state, the same condition observed during hypobaric hypoxia exposure, is found in association with many diseases affecting the respiratory and cardiovascular systems. However, a similar condition is observed nonpathologically in individuals when going to high altitudes. The possibility of studying hypoxia in a nonpathological context, within a controlled environment, has represented an incredible opportunity for understanding the more intimate mechanisms primed by the condition and the metabolic rearrangements occurring in the body. Heretofore, very few studies have had access to such controlled conditions to examine this process.

Our study, based on NMR Spectroscopy, identified potential alterations of metabolic profiles while also attempting to distinguish between early and late adaptations. Nevertheless, defining the threshold beyond which one can begin to speak of chronic hypoxia and record its consequences seems rather elusive. From multivariate analyses (both in PCA and PLS-DA models, obtained for the whole data), a time-dependent metabolomic change was clearly observed. The most conspicuous variation was observed for the T0 time point with respect to the others. This establishes the existence of a rapid hypoxic adaptation well away from the basal conditions. Subsequently, the differences found between the TII-TIII and the TIV-TVI clusters indicate that between 30 and 90 days of exposure to hypoxic conditions, there is perhaps a deeper metabolic switch, signaling a transition from an early to a late adaptive response. Several metabolic adaptations are implemented by the organism to preserve the oxygenation of tissues and organs, showing, accordingly, a rather variable tolerance to hypoxia. This is achieved through a set of extremely rapid adaptive physiological reactions in the acute phase, which, as the condition persists, is followed by more complex mechanisms in the long run.

Cells mainly depend on glucose and O_2_ for energy generation even if they can take advantage of other carbon sources such as amino acids and fatty acids (FAs). A full utilization of the glucose molecule for energy production occurs through glycolysis, the tricarboxylic acid (TCA) cycle, and electron transport chain (ETC), ultimately going on to generate ATP. In hypoxic conditions, protein leakage in the ETC impair the OXPHOS pathway, leading to an increase in ROS levels and to a reduction in the ATP/ADP ratio, which subsequently cause a reprogramming of the cellular metabolism driven by HIF transcription factors [21]. This brings an increase in glucose uptake, glycolytic activity, and gluconeogenesis processes to sustain energy production [8]. The TCA cycle, therefore, allows other substrates to be metabolized as carbon sources through anaplerotic processes. The most prominent alternative substrate to glucose is glutamine, circulating in the blood with a concentration of ~500–800 μM [22,23].

Our assays detected an increase in the circulating levels of glutamine (Figure 2A) following the arrival of the subjects at Concordia base and, at the same time, a reduction in circulating glutamate (Figure 2B). Moreover, the metabolic analysis found a perturbation in the glutamine glutamate synthetic pathway in the comparison of T0 vs. TI and TI vs. TVI (Figure 4A,B). During hypoxia, skeletal muscle volume, which constitutes almost 40% of the organism mass, goes through a remodeling because of an alteration in protein synthesis/degradation ratio coupled with a metabolic change [24]. This process allows for the release of substrates that are used by the body as fuel, as well as glutamine; the organ is indeed the main site of production and storage of the amino acid [23,25]. However, Chicco et al. have observed that, especially for oxidative muscle fibers, the rearrangement in some cases is not as profound as has been long assumed but is rather a substitution of some elements and enzymes with the purpose of optimizing and sparing O_2_ consumption [26]. Recently, it has been shown that such processes also involve other regions of the body; the optimization is performed mainly by substituting subunits of the mitochondrial ETC with others more efficient in the use of O_2_, a phenomenon that mainly involves complex I and IV [27]. To further supply the metabolic needs of the hypoxic body, glutamine represents an important gluconeogenic substrate for liver and kidneys, with the latter capable of contributing 25% of the energy demand of the organism [28].

The anabolic processes in hypoxia are also modulated through an increased activity of pyruvate dehydrogenase kinase (PDDK1), which inhibits pyruvate dehydrogenase (PDH) by preventing the conversion of pyruvate to acetyl-CoA. The net result is a decreased influx of substrates to the TCA cycle, a slowing down of OXPHOS, and a detour of cellular metabolism toward an increased conversion of pyruvate to lactate by lactic dehydrogenase (LDH) [29]. Consequently, FAs have an extremely important role in feeding the TCA cycle. Indeed, an increase in their synthesis is observed, mainly by liver, skeletal muscle, and adipocytes, as is their uptake in numerous parts of the body [30]. What we observed supports the hypothesis that glutamine and glutamate are among the main lipogenic substrates during hypoxia, which are converted by reductive carboxylation to α-ketoglutarate [31]. This process appears to play an important role at the neuronal level, with evidence showing FA levels increasing due to a reduction in their oxidation at the expense of the synthesis processes. This also reaffirms Broose et al. observations, who showed a neuronal reduction in the incorporation of aspartate and glucose at the expense of glutamine and, secondarily, glutamate into newly formed lipid molecules [32]. The increase in circulating lipids levels, found in the winter-over samplings (Figure 2C), supports the hypothesis of an increased lipogenesis and centrifugal mobilization during hypoxia. It has been observed that HIF-1α increases the expression of receptors for LDL and VLDL, especially at the level of cardiac muscles [33]. The heart is an energy-intensive organ with a great capacity to exploit different substrates to maintain its activity and protect itself from ischemic damages. A higher FA cardiac content has been found in individuals adapted to living at high altitudes [34]; nevertheless, organs suffering chronic hypoxic conditions tend to favor glycolysis to satisfy their energy requirements, downregulating β-oxidation. The reason for this lies in the significant O_2_ expenditure required by the enzymes involved in the process [22,35]. Therefore, FAs tend to be accumulated in the form of triglycerides and lipid droplets as a form of energy storage [36].

Although β-oxidation is downregulated, a small amount of O_2_ allows FAs to be converted to acetyl-CoA, which enters the TCA coupled with glycolytic oxaloacetate. In states of reduced intracellular glucose concentration, oxaloacetate levels drop because of its redirection to the gluconeogenesis pathway. This condition, when coupled with an overproduction of acetyl-CoA, further shifts the metabolism of acetyl-CoA toward the formation of ketone bodies and specifically acetone and β-hydroxybutyric acid [37,38]. The latter, according to pairwise OPLS-DA in this study, showed an increase in its levels in the comparison of T0 with TVI and TI with TVI (Figure 3B,C). Consequently, the production of ketone bodies appears to occur more delayed in time and is among the later adaptive responses deployed by the organism toward hypoxia.

The observed reduction in the circulating glutamate levels (Figure 2B) could be justified in different ways. This amino acid holds considerable importance during hypoxic conditions and can be imported into the cell or generated from glutamine with the production of an ammonium ion. It is indeed involved in some major synthetic pathways, as it can enter the TCA cycle after being converted to α-ketoglutarate to contribute to FA synthesis and intermediate formation by glutamate dehydrogenase, leading to the generation of an ammonium ion. Alternatively, the formation of α-ketoglutarate may result from the activity of glutamic-oxaloacetic transaminases or glutamic-pyruvate aminotransferases, with the production of the nonessential amino acid asparagine, from which the nonessential amino acids lysine, threonine, and methionine are derived [39]. The reduced O_2_ amount and the associated increase in ROS is countered by the cell through enzymatic and nonenzymatic mechanisms. Aminothiols are the major nonenzymatic antioxidant compounds that directly quench ROS, the most abundant of which is reduced glutathione (GSH) [40]. Glutathione is a tripeptide consisting of glutamate, cysteine, and glycine; glutamine therefore constitutes the first amino acid required for its production but can also serve as a precursor for glycine synthesis. Following ROS neutralization, oxidized glutathione (GSSG) is converted back to GSH by the oxidation of NADPH to NADP^+^ [41].

NMR analysis also showed a reduction of GlycA, which represents a pool of proteins of the acute phase of the inflammatory response, whose release is modulated predominantly at a hepatic level under the stimulus of proinflammatory cytokines such as IL-6. This heterogenous set of proteins includes mainly α1-acid glycoprotein, haptoglobin, α1-antitrypsin, α1-antichymotrypsin, and transferrin [42]. They are characterized by the presence of a series of N-linked acetylglucosamine (GlcNAc) molecules on their structure, assembled from the UDP-GlcNAc derived from the hexosamine cycle [43]. The GlycA levels are considered a powerful indicator of the systemic inflammatory condition, more effective than the C-reactive protein level [44]. The hypoxic condition is generally associated with an inflammatory process and stress condition, mainly related to ROS increase. The decline in the levels of GlycA compared with the baseline value was evident from the TI, and gradually decreased relative to earlier samplings (TI-TIII); then, it remained roughly unvaried relative to the later time points (TIV-TVI) (Figure 2D). Glycosylation is an essential process for the maintenance of proteins structure and function and can modulate their localization. This process is handled by glycosyltransferases from nucleotide sugars, of which glucose is the main precursor. Physiologically, it is estimated that between 1 and 3% of the cell glucose is diverted to this pathway [43]. However, HIF-mediated metabolic rearrangement prioritizes the targeting of glucose to the glycolytic process for ATP production at the expense of other pathways. At this point, the glycosylation process slows down and is fed through other sources, first and foremost glutamine, to avoid the onset of endoplasmic reticulum stress [45]. At the same time, changes in the activity of glycosidases and glycosyltransferases promote alterations in the glycosylation pathway of several proteins. If O-glycosylation occurs during hypoxia, an increased activity and/or expression of O-GlcNAc transferase is observed at the expense of O-GlcNAcases, but the reason for the alteration in the N-glycosylation pathway remains to be elucidated [46].

To the best of our knowledge, NMR studies on the use of GlycA as a marker of inflammation in hypobaric hypoxia are limited. What has been observed suggests that the parameter we measured may be strongly influenced by metabolic alterations in the glycosylation pathway and not by the absence of inflammatory phenomena. Therefore, a further investigation in this regard is needed.

Metabolic pathway analysis (MetPA) identified several pathways that were disrupted during the period of hypoxia. Alterations in these pathways involving glutamine and glutamate, observed by comparing T0 with TI and TVI, are explained above.

A perturbation in the arginine biosynthetic pathway was found between T0 and TI, but also between T0 and TVI (Figure 4A,B). Under hypoxic conditions, one of the responses set in motion by the body is aimed at increasing the blood perfusion of organs which occurs after an initial vasoconstrictive phase intended to protect major areas to the detriment of peripheral ones [47,48]. This is achieved through a systemic vasodilatory response which is followed by vasculogenic processes. Nitric oxide (NO) is a molecule with potent vasodilatory functions, has arginine as precursor, and is synthesized in the body by NO synthase (NOS), for which at least three isoforms are currently known [49]. The fact that the pathway appears altered even in the comparison of T0 with TI indicates that its activation is extremely rapid and related to the acute acclimatization process. Meanwhile, the pathway disruption observed between T0 and TVI is indicative of its activation even at the time when the acclimatization should have peaked—the end of the winter-over.

Finally, we observed an alteration in the phenylalanine metabolism and phenylalanine, tyrosine, and tryptophan biosynthesis pathways between TI and TVI (Figure 4C). Many amino acids play a role in modulating attitudes and moods. Some, including glutamate, glutamine, aspartate, serine, glycine, threonine, alanine, and histidine, are related to the onset of depressive disorders. Tryptophan is a common precursor to the formation of serotonin and kynurenine [50], while phenylalanine and its derivate tyrosine are precursors of dopamine, norepinephrine, and epinephrine. These neurotransmitters take part in several physiological processes and modulate the behavioral and emotional sphere [51,52,53,54]. This observation may explain a decline in mood and cognitive performance, events already observed in association with circumstances experienced during the winter-over. The prolonged distance from families, the inability to freely leave the base, and the alteration of circadian rhythms, all find a metabolic correlation in our study.

In conclusion, the NMR metabolomics approach of the blood sera of the participants involved in the study returned considerable information of interest. Some of the obtained data made it possible to reinforce knowledge already acquired in the context of hypoxia research, in which it is often difficult to study the condition for prolonged durations in a manner that was done in our study. In addition, the samples we analyzed derive from subjects observed under ideal conditions for the study of hypoxia adaptation and in the absence of perturbing measurement elements or pathologies.

Unfortunately, a limitation of the study remains the absence of measurements after the return from the Antarctic continent, made impossible due to concomitant COVID-19 pandemic outbreak. Such measurements would have provided important information on the changes that occur after a prolonged period of hypoxia and would have helped determine whether a complete return to baseline parameters prior to departure takes place after such a long period of exposure to such conditions.

## 4. Materials and Methods

### 4.1. Participants

The study was performed on eight healthy men, with ages between 25 and 55 years, all of whom were crew members of the Concordia Station Antarctic base and part of the European Space Agency (ESA) Life Science campaign (XV winter-over, 2019). All individuals provided informed consent to the conduct of the study and the collection of blood samples at selected time points, and the study was approved by the Ethical Committee at “San Paolo” Hospital (Milan, Italy).

### 4.2. Biological Sample Gathering

Study subjects underwent peripheral blood collection from the antecubital vein. All the samplings were performed in the early morning in fasting conditions. The blood was collected in a dedicated anticoagulant-free tube and centrifuged in a swing-out head centrifuge for 10 min at 800 RCF. The sera obtained were aliquoted and immediately stored at a temperature ranging from −50 °C to −80 °C until analysis. Since the base lacks −80 °C refrigerators and liquid nitrogen tanks, sample storage was carried out in the dug-out structure below the base dedicated to the deposition and preservation of ice cores. This environment ensures a constant temperature of at least −50 °C regardless of the external conditions. Further storage was attained at −80 °C.

An initial sampling session (T0) was conducted at the ESA Centre in Cologne (Germany, 91 m ASL) to obtain the baseline value of each parameter. At the Concordia base, blood collection was performed within the first 48 h of arrival (TI). Subsequent samplings occurred after 7 (TII), 14 (TIII), 90 (TIV), 180 (TV), and 270 (TVI) days after arrival. The ESA base physician monitored, throughout the stay, the health condition of each subject involved in the study. No occurrence of “winter-over syndrome” or onset of noteworthy illnesses was reported.

### 4.3. Sera Preparation for the Metabolomic Analysis

The frozen samples, described with the available subject data in Appendix A, were thawed at room temperature prior to NMR analysis. Briefly, ~150 μL of serum sample was mixed with 400 μL of saline buffer solution at pH 7.4 (NaCl 0.9%, 50 mM of sodium phosphate buffer in D_2_O containing trimethylsilyl propionic-2,2,3,3-*d*_4_ acid sodium salt, TSP), to minimize the pH variation and then transferred to a 5 mm NMR tube. The NMR experiments were recorded on a Bruker Avance III NMR spectrometer (Bruker, Ettlingen, Germany) operating at 600.13 MHz for ^1^H observation, equipped with a TCI cryoprobe (Triple Resonance inverse Cryoprobe), and incorporating a *z*-axis gradient coil and automatic tuning-matching. Experiments were completed at 300 K in automation mode after individual samples were loaded on a Bruker Automatic Sample Changer, which was interfaced with the software IconNMR version 5.0.12 (Bruker). For each sample, a standard 1D ^1^H one-dimensional spectrum (with suppression of water and broad protein resonances) Carr–Purcell–Meiboom–Gill (CMPG) spin-echo sequence was acquired with 128 transients, 16 dummy scans, a 5 s relaxation delay, an FID (free induction decay) size of 64 K data points, a spectral width of 12,019.230 Hz (20.0276 ppm), an acquisition time of 1.36 s, a delay of 1.2 ms (d20, fixed echo time to allow elimination of J modulation effects according to the Bruker pulse program code, with a loop of 126 cycles for T2 filter), a total value of total spin–spin relaxation delay of 302.4 ms, and solvent signal saturation during the relaxation delay. The resulting FIDs were multiplied by an exponential weighting function corresponding to a line broadening of 0.3 Hz before Fourier transformation, automated phasing, and baseline correction. Moreover, 2D NMR spectra (^1^H Jres, ^1^H-^1^H COSY, ^1^H-^13^C HSQC and ^1^H-^13^C HMBC) were also randomly acquired for assignment purposes, together with a comparison with published data and public databases [19,55]. The NMR spectra were processed using Topspin 3.6.1 and Amix 3.9.13 (Bruker, Biospin, Italy), both for simultaneous visual inspection and the successive bucketing process for multivariate statistical analyses.

### 4.4. Multivariate Analysis of NMR Data

The entire NMR spectrum (in the range 9.0–0.94 ppm) was segmented in fixed rectangular buckets 0.02 ppm in width and successively integrated. Chemical shifts were referenced to the internal standard (TSP, δ_H_ 0.00 ppm), and the spectral regions between 5.20–4.23 ppm were discarded to avoid the effects of water suppression. Moreover, the spectral regions between 3.65–3.60 and 1.20–1.10 ppm, due to the residual peaks of solvents and contaminants (disinfection material), were not taken into consideration. The total sum normalization and the Pareto scaling procedure (performed by dividing the mean-centered data by the square root of the standard deviation) were applied to minimize small differences due to sample concentration and/or experimental conditions among samples [56]. The input data matrix, made by the NMR descriptors (the buckets), were labelled with the central chemical shift value for the specific 0.02 ppm width. Each bin was normalized to a total spectral area and then mean-centered in order to allow for the comparison between metabolic variations. Multivariate statistical analysis, unsupervised principal component analysis (PCA), supervised partial least squares discriminant analysis (PLS-DA), and supervised orthogonal partial least squares discriminant analysis (OPLS-DA) were performed to examine the intrinsic variation in the data using free-source MetaboAnalyst 5.0 software (https://www.metaboanalyst.ca/, accessed on 8 May 2023) [57]. The quality of the statistical models was validated using R^2^ and Q^2^ parameters, which respectively indicated the goodness of fit and quality of the prediction. A random permutation test, set to 100, was also performed, with empirical *p*-values of Q^2^ and R^2^Y being < 0.01 (0/100) and thus statistically valid. The unbiased NMR bins corresponded to high VIP (variable importance in projection) values >2, and a weighted sum of squares of the OPLS-DA weight was considered for discrimination purposes and further assigned to corresponding metabolites. The significant differences of the mean values for discriminant metabolites were calculated with a classical statistical analysis (one-way ANOVA) following Tukey’s honestly significant differences (HSD) post hoc test. The level of statistical significance was considered to be a *p*-value < 0.05 with a 95% confidence interval. The box plots of the metabolites responsible for the group clustering are shown among the compared groups. Moreover, a repeated measures one-way ANOVA was conducted on individuals to examine the effect of perturbations that were potentially related to the physiological adaptation on response time by using R Statistical Software (v4.1.2; R Core Team 2021). Y-axes are represented as relative units. Data were mean-centered and normalized to the total spectral area. Due to mean-centering and normalization process, we obtained a negative scale in the *Y*-axis of the bins (Metaboanalyst program analysis) [56,58,59,60]. In order to directly measure the concentration of blood glucose, the ERETIC 2 (electronic reference to access in vivo concentrations) methodology was also used as a quantification experiment on the NMR spectra. An internal reference compound was used (solution of gallic acid in D_2_O), and the specific proton resonance at 7.05 ppm (s, 2H) was considered for the quantification analysis. Peak integration, ERETIC measurements, and spectra calibration were obtained by using specific subroutines of Bruker Top-Spin 3.6.1 software [61].

### 4.5. Metabolic Pathway Analysis

The metabolic pathways analysis was performed using the significant metabolites identified in each of the previous supervised models OPLS-DA with MetaboAnalyst 5.0 software. Metabolites of interest were selected by distinctive unbiased NMR bins (VIP > 2), identified in the corresponding VIP plots, and used as the input matrix for the metabolic pathway analysis. The software identified the most significantly perturbed pathways using the KEGG PATHWAY (http://www.genome.jp/kegg/, accessed on 8 May 2023) metabolic database. Pathway impact was calculated as the sum of the importance measures of the matched metabolites normalized by the sum of the importance measures of all metabolites in each pathway [57,59,62].

### 4.6. Evaluation of Glucose Levels

The glucose content in each serum sample was evaluated colorimetrically by using a CBA086 Glucose Assay Kit according to the manufacturer’s instructions (Merk, Germany). The assay was set up by combining the two kit reagents in the following proportions: 4 mL of R1 (phosphate buffer pH 7.4 100 mmol/L; phenol 9 mmol/L; glucose oxidase ≥ 15,000 U/L; peroxidase ≥ 1200 U/L) and 1 mL of R2 (4-aminophenazolone 80 mmol/L). A volume of 10 μL of sample was added to 1000 μL of R1+R2 mix. In parallel, a blank and a control were prepared by adding 10 μL of distilled H_2_O or 10 μL of standard, glucose 100 mg/dL (5.55 mmol/L) (provided with the kit), respectively, to 1000 μL of mix. The sample was shaken and incubated for 10 min at 37 °C and then read at 520 nm using a spectrophotometer (Bio-rad, Hercules, CA, USA). The glucose concentration was then obtained by relating the extinction values of sample (EC) and standard (ESTD) against the reagent blank using the following formula:Glucose concentration (mg/dL) = (EC/ESTD) × Standard concentration

Data representation (Appendix A) and analysis were performed with Prism 9.0 software (GraphPad Software, Inc., La Jolla, CA, USA). Results are presented as mean ± SEM. Data were processed using one-way ANOVA, which was followed by Dunnett’s post hoc test. Statistical analyses were performed, and a *p*-value < 0.05 was used as the level of significance.

## Figures and Tables

**Figure 1 ijms-24-08459-f001:**
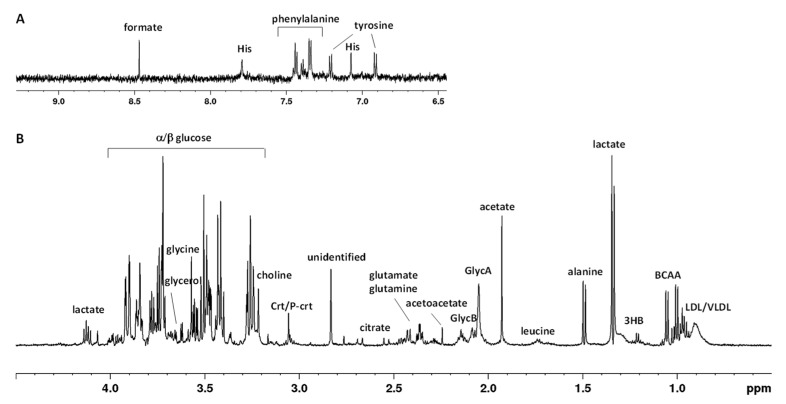
Typical 600 MHz ^1^H CPMG (Carr–Purcell–Meiboom–Gill) NMR spectrum of human serum in the aromatic (**A**) and aliphatic (**B**) regions. Abbreviations used: BCCA—branched chain amino acids (valine, isoleucine, leucine); 3HB—3-hydroxybutyrate; His—histidine; LDL/VLDL—low-/very-low-density lipoprotein; GlycA/GlycB—N-acetyl glycoprotein; Crt/P-crt—creatine/phosphocreatine.

**Figure 2 ijms-24-08459-f002:**
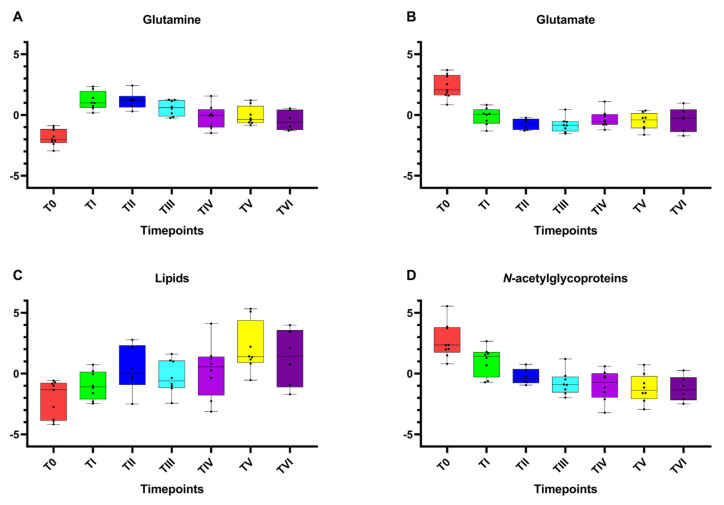
Box and whisker plots of the significantly altered metabolites (spectral bins) according to NMR data analysis. Y axes are represented as relative units. Data were mean-centered and normalized to the total spectral area. Due to the mean-centering and normalization process, we obtained a negative scale in the *Y*-axis of the bins (Metaboanalyst program analysis). The bar plots show the normalized values (mean +/− standard deviation). The boxes range from the 25% and the 75% percentiles; the 5% and 95% percentiles are indicated as error bars; single data points are indicated by black dots. The variations between the timepoints considered of (**A**) glutamine, (**B**) glutamate, (**C**) lipids, and (**D**) *N*-acetylglycoproteins are shown.

**Figure 3 ijms-24-08459-f003:**
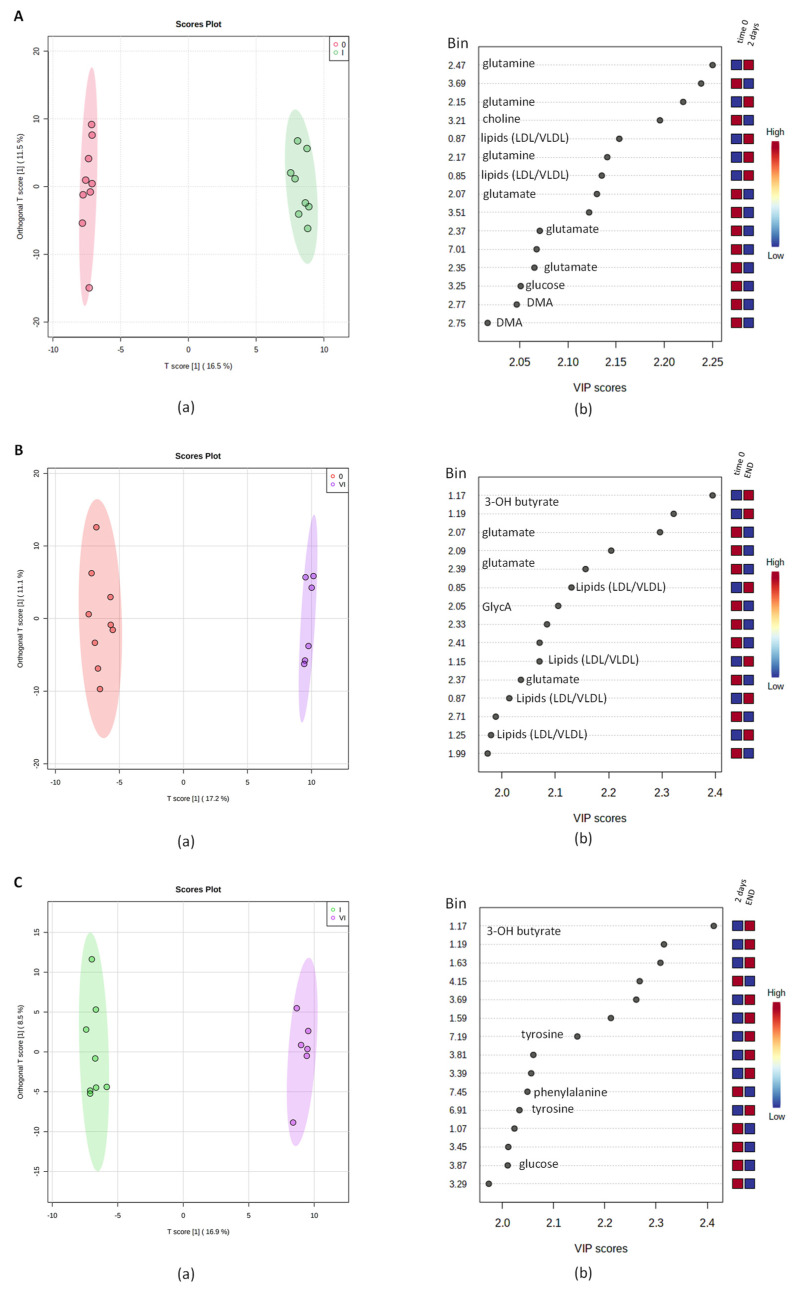
Pairwise OPLS-DA models scores plots (a) and VIP scores for metabolites related to buckets with different relative concentrations among the two considered groups (b), obtained for predeparture (T0) vs. TI (**A**), predeparture (T0) vs. TVI (**B**) and TI vs. TVI (**C**) time points.

**Figure 4 ijms-24-08459-f004:**
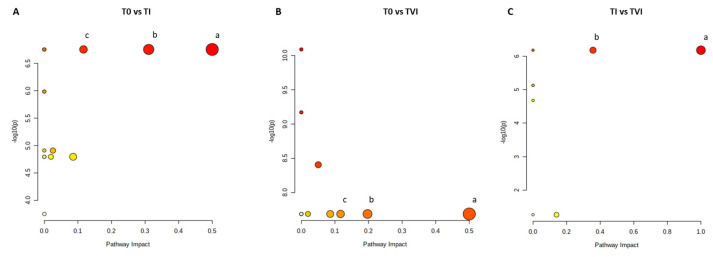
Metabolic pathways. Color intensity (white to red) reflects increasing statistical significance, and circle diameter varies with the pathway impact. The graph was obtained by plotting on the y-axis the −log10 transforms of *p*-values from the pathway enrichment analysis and on the x-axis the pathway impact values derived from the pathway topology analysis. The most significant pathways (*p*-value < 0.05; pathway impact > 0.1) are depicted (**A**) (a) _D_-glutamine and _D_-glutamate metabolism; (b) alanine, aspartate, and glutamate metabolism; (c) arginine biosynthesis. (**B**) (a) _D_-glutamine and _D_-glutamate metabolism; (b) alanine, aspartate, and glutamate metabolism; (c) arginine biosynthesis. (**C**): (a) phenylalanine, tyrosine, and tryptophan biosynthesis; (b) phenylalanine metabolism.

## Data Availability

Data availability is restricted in order to safeguard the privacy of the study subjects.

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
