# Peer review of "Blood Metabolite Profiling of Antarctic Expedition Members: An 1H NMR Spectroscopy-Based Study"

_ijms, 2023, doi:10.3390/ijms24098459_

Round 1

Reviewer 1 Report

Review of the paper:

Blood metabolite profiling of Antarctic expedition members: an 1H NMR spectroscopy-based study

Laura Del Coco, Marco Greco, Alessandra Inguscio, Anas Munir, Antonio Danieli, Luca Cossa, Maria Rosaria Coscia, Francesco Paolo Fanizzi, Michele Maffia

The authors propose to study, using proton NMR-based metabolomics of serum, the acute and chronic impact of exposition to hypoxia during a 9 months Antarctic expedition, during which individuals were exposed to atmospheric pressure identical to high altitude (4800 m). During the study, the authors gathered blood samples of 8 individuals at 7 time points, one before the expedition and 6 from their arrival to the Antarctic base to the end of the expedition. They have identified time dependant metabolic changes, mainly of 2 metabolites: glutamate and glutamine, and of lipids and N-acetylglycoproteins.

The paper is well written and the introduction clearly sets the context and interest of the study.

However, I have several concerns with the methods section, which greatly affects the reliability of the results presented:

- first, some points are not sufficiently detailed, see points 1, 2, 5.

- Secondly, the negative amplitude of metabolites, presented figure 2, is not possible with CPMG pulse sequence, and “normalisation to sum”, see point 4 below, cannot produce negative amplitudes. Negative amplitudes are obtained with CPMG when the inter-pulse delay (delay between the 180° pulses) is not adequately chosen. It must therefore be changed until all peaks can been positively phased. This greatly affect the reliability of the whole paper since the amplitude of metabolites in NMR spectra are the main results presented and the basis of the discussion.

As a consequence, even if the discussion is well written and is interesting, it must be re-considered after consolidating the results.

Material and methods

1/ Line 440 : total spin-spin relaxation delay of 1.2ms: seems very low, is it the total echo time of the CPMG? Usually 80 or 120 ms are used with serum samples. The important attenuation of lipid resonances is not consistent with this delay of 1.2 ms.

2/ Sera preparation: Was blood sampled in fasting people? This must be clearly indicated since food habits have probably drastically changed between T0 and the post-arrival at Concordia base.

Was the time elapsed between blood sampling and serum freezing short (less than 2h) and identical for all subjects/all time points?

3/ Univariate statistical analysis: since blood is collected at seven time points in the same subjects, this should be taken into account, by using repeated measures ANOVA

4/ Line 475 and legend fig 2: NMR always produce positive peaks, i.e. the phasing of the spectra must be correctly performed so as all peaks are positive, and the base line must be corrected and put to zero before bucketing the spectra. As a result, the normalisation procedure, consisting of dividing the value of each bucket, positive, by the total area, positive, produce positive values and cannot produce negative peaks. This is completely not understandable why almost all metabolites figure 2 have negative amplitudes.

5/ Bucketing line 451: each bucket contains only a part of the peaks of a given metabolite, e.g. the peaks at 2.35 and 2.44 ppm for glutamate and glutamine, respectively. So what does the amplitude of glutamate and glutamine correspond to? The sum of buckets? this should be indicated, as well as the peaks, or the spectral region, used for univariate statistics for all metabolites tested, at least as supplemental data.

6/ Line 455: total sum normalisation: Glucose peaks are very intense compared to most of the metabolites, and they are numerous. As a consequence, the weight of glucose bins when using total sum normalisation is very important, and this can mask small variation of other metabolites. Their exclusion during the total sum normalisation procedure should be tested, even more if glucose exhibits variability between spectra.

7/ The number of metabolites detected is low, e.g pyruvate should be detected at 2.37 ppm.

8/ Creatine and phosphocreatine give two different peaks around 3.05 ppm

9/ Glucose could have been quantified in NMR spectra and its amplitude compared between different time points.

Results

Glucose level: high inter-individual variability observed, had subjects underwent a 12 h fasting before blood sampling? This should be indicated

Is this variability also observed for glucose peaks in NMR spectra?

Reviewer 2 Report

The manuscript “Blood metabolite profiling of Antarctic expedition members: an 1H NMR spectroscopy-based study” [ijms-2341183-peer-review-v1] written by Laura Del Coco, Marco Greco, Alessandra Inguscio, Anas Munir, Antonio Danieli, Luca Cossa, Maria Rosaria Coscia, Francesco Paolo Fanizzi and Michele Maffia describes a detailed 1H NMR based metabolic study of blood sera taken from humans in a highland in polar environment. The data gained are further analyzed by a principal component analysis and multivariant data analysis. The results are in particular used to identify stress condition indicators and pro-inflammatory marker.   

The reviewer has expertise in the molecular field of structure determination (in particular NMR) and hence mostly refers to this part of the manuscript with the review.

The overall work seems to be quite well planned and performed. The practical investigation has been made carefully with state of the art methods. The reported results sound perspicuous. The further analysis of the data as well as the discussion and conclusion are sensible and detailed with respect to entire data gained. However, there are some points listed below, which should be further addressed by the author.

The manuscript is hence of interest in the fields of Spectroscopy, Biochemistry, and Medicinal Chemistry. It is worth publishing in general. However, some details of the NMR measurements should be discussed in some detail with respect to possible errors and the presentation should be improved. Hence, there are a few comments listed below, which should be taken into account by the authors prior to acceptance of the manuscript. Therefore, the manuscript is not yet in a form to be published in “International Journal of Molecular Science“.

General Comments:

1)
The authors use a spin echo sequence in an automated setting to record the 1H NMR spectra. For this purpose, the authors should discuss more critically what effects this can have on the quantifying evaluation of the gained 1H NMR data. In doing so, authors should consider the following points:
1a) The spectra are recorded in water, whereby some of the components investigated are only moderately water-soluble (e.g. LDL/VLDL) and can form aggregates in the investigated samples. Or they are quite large molecules (glycoproteins). Even using a quite short relaxation delay (1.2 ms), this leads to differently intensive T2 relaxation for the various components examined, depending on their solubility. And consequently to different signal intensities in the 1H NMR spectra.
1b) Line broadening in baseline correction, especially in automated mode, can lead to slightly different results influenced by "impurities" or (suppressed) solvent signals. Also this can lead to differences in the integrals across individual buckets (or signals).
From his own experience, the reviewer estimates the errors that result from this in relation to the determination of the integrals to be at least 10-15%.

2) It is not entirely clear to the reviewer which buckets were used in detail for quantifying the individual compounds. The authors are encouraged to describe this in a little more detail. Is there a reason why fixed and not variable buckets have been chosen?

3)
The assignment of the individual 1H NMR signals to the various components examined appears to be correct to the reviewer. However, this cannot be clearly derived from the HSQC in the Supplementary Material. The authors are therefore encouraged to show a few more of their recorded 2D NMR spectra there and to make the interpretation comprehensible.

4) What is meant by "pre-processing" (line 140) in detail?

Minor Comments:

5)
Why hasn't Glc been quantified using the NMR method?

6) Small caps should be used to indicate the absolute configuration in Fischer-nomenclature of carbohydrates “D” (e.g. line 206 and elsewhere in the manuscript).

7) For “liter” always capital “L” should be used (e.g. in line 218).

8) The reviewer assumes "GlcNac" means "GlcNAc"!? (e.g. lines 353 und 354)

9)
"N" as prefix for "nitrogen" should be written in italics. (e.g. line 138: N-acetyl glycoproteins)

10 Line 413 far right: Is that a sign for degrees "°"?

11) line 429: “
trimethylsilyl propionic-2,2,3,3-d 4 acid sodium salt” “d” in italics and the following “4” in subscript.

12)
Authors are encouraged to check that the journal's guidelines have been followed for all references. Here the reviewer noticed a few small irregularities in the presentation. (e.g. journal names in italics or not)

Round 2

Reviewer 1 Report

Second review of the paper:
Blood metabolite profiling of Antarctic expedition members: an 1H NMR spectroscopy-based study
Laura Del Coco, Marco Greco, Alessandra Inguscio, Anas Munir, Antonio Danieli, Luca Cossa, Maria Rosaria Coscia, Francesco Paolo Fanizzi, Michele Maffia

The paper has been greatly improved, only some minor points remain to be addressed, i.e.:
- Material and method
1/ the total echo time is an important parameter, as important as or even more important than the D20 interpulse delay, so it must be mentioned in the paper.
4/ I apologize for my misunderstanding, mean-centering for data visualization is not current with NMR data, but has the advantage of allowing comparison between all metabolic variations. So yes, I agree that the mean-centering process can produce negative values, but not the normalization to total spectral area, this latter being performed before mean-centering, at the end of the bining process. The Y axes should all be identical (min, max) for a better comparison of all metabolic variations.
I therefore suggest something like that “each bin was normalized to total spectral area then mean-centered in order to allow comparison between metabolic variations”

Author Response

 Second review of the paper: Blood metabolite profiling of Antarctic expedition members: an 1H NMR spectroscopy-based study Laura Del Coco, Marco Greco, Alessandra Inguscio, Anas Munir, Antonio Danieli, Luca Cossa, Debora Musarò, Maria Rosaria Coscia, Francesco Paolo Fanizzi, Michele Maffia The paper has been greatly improved, only some minor points remain to be addressed, i.e.: - Material and method 1/ Total echo time is an important parameter, as important as or even more important than the D20 interpulse delay, so it must be mentioned in the paper. 

Answer: according to the referee’suggestions, the total echo time has been reported in the revised manuscript, lines 449-452. 

4/ I apologize for my misunderstanding, mean-centering for data visualization is not current with NMR data, but has the advantage of allowing comparison between all metabolic variations. So yes, I agree that the mean-centering process can produces negative values, but not the normalization to total spectral area, this latter being performed before mean-centering, at the end of the bining process. The Y axes should all be identical (min, max) for a better comparison of all metabolic variations. I therefore suggest something like that “each bin was normalized to total spectral area then meancentered in order to allow comparison between metabolic variations” 

Answer: according to the referee’suggestions, the original figure has been modified in order to better highlight the position of overall mean-centered variations. Moreover, the sentence has been reported in the revised manuscript, lines 471-473. 

Some typos have been corrected through the revised manuscript 
